# Microbiota Dysbiosis in Parkinson Disease—In Search of a Biomarker

**DOI:** 10.3390/biomedicines10092057

**Published:** 2022-08-23

**Authors:** Julia Maya Nowak, Mateusz Kopczyński, Andrzej Friedman, Dariusz Koziorowski, Monika Figura

**Affiliations:** 1Student Scientific Group, Department of Neurology, Faculty of Health Sciences, Medical University of Warsaw, 02-091 Warsaw, Poland; 2Department of Neurology, Faculty of Health Sciences, 02-091 Warsaw, Poland

**Keywords:** Parkinson, neuroinflammation, gastrointestinal tract, microbiota, intestine

## Abstract

Numerous studies have highlighted the role of the gastrointestinal system in Parkinson disease pathogenesis. It is likely triggered by proinflammatory markers produced by specific gut bacteria. This review’s aim is to identify gut bacterial biomarkers of Parkinson disease. A comprehensive search for original research papers on gut microbiota composition in Parkinson disease was conducted using the PubMed, Embase, and Scopus databases. Research papers on intestinal permeability, nasal and oral microbiomes, and interventional studies were excluded. The yielded results were categorized into four groups: Parkinson disease vs. healthy controls; disease severity; non-motor symptoms; and clinical phenotypes. This review was conducted in accordance with the PRISMA 2020 statement. A total of 51 studies met the eligibility criteria. In the Parkinson disease vs. healthy controls group, 22 bacteria were deemed potentially important. In the disease severity category, two bacteria were distinguished. In the non-motor symptoms and clinical phenotypes categories, no distinct pathogen was identified. The studies in this review report bacteria of varying taxonomic levels, which prevents the authors from reaching a clear conclusion. Future research should follow a unified methodology in order to identify potential biomarkers for Parkinson disease.

## 1. Introduction

Parkinson disease (PD), a neurodegenerative disorder characterized by gradual dopaminergic neuronal loss in the substantia nigra (SN), is most recognizable for its hallmark motor manifestation: tremor, rigidity, and bradykinesia. However, PD also involves a wide range of less distinctive, but quite common non-motor symptoms (NMS) [1,2,3]. They are often neglected during patient evaluation, since movement impairment and its eventual progression tend to be the most pronounced aspects of PD [4]. Nevertheless, the non-motor manifestation of this disease has been able to provide some valuable insight into its possible pathogenesis. Hyposmia and constipation are particularly notable, as they may precede motor symptom onset by even 20 years [5]. This observation suggests a peripheral origin of PD. Additionally, the discovery of Lewy bodies (abnormal deposits of alpha-synuclein) in the intestinal submucosal and myenteric plexuses of PD patients explicitly highlights the role of the intestine in PD etiology [6,7,8]. Dual-hit theory, proposed by Hawkes, Tredici, and Braak, implicates two places in the body as sites of PD origin—the olfactory nerves and intestinal plexuses [9]. Moreover, it suggests that neural damage initiated in the gut spreads to the central nervous system (CNS) via the vagus nerve [9], and it is only after neurodegeneration reaches the substantia nigra, the motor phase of PD begins [5].

A recent paper by Horsager et al. proposed a new and updated model of PD pathogenesis [10]. It categorizes PD into two subtypes: the body—first phenotype, and the brain—first phenotype [10]. Clinical [11,12] and neuropathological [13,14] evidence suggests that, in a subset of PD patients, neurodegeneration originates and firstly propagates through the CNS, with the autonomic nervous system being affected at a later stage of this disease [10]. These findings are inconsistent with the above-mentioned dual-hit hypothesis, hence the proposal of distinguishing a separate phenotype of PD—the brain-first subtype [9]. Nevertheless, PD patients categorized into the body-first subtype exhibit symptoms in an order compatible with the dual-hit hypothesis, and it is this particular group that pertains to this systematic review [9].

To date, the exact trigger of intestinal neurodegeneration remains elusive. However, the process of alpha synuclein aggregation, and thus Lewy body formation, in the intestine correlates with increased intestinal permeability, which was demonstrated by a study comparing PD subjects to healthy controls [15]. This dysfunction in intestinal barrier integrity is most likely initiated by proinflammatory factors produced by certain bacteria residing in the gut [15]. A recent study found a significant correlation between the consumption of narrow spectrum penicillin and a higher prevalence of PD [16]. This further supports the theory that gut dysbiosis may trigger intestinal neurodegeneration, as it is well-documented that antibiotic exposure has a selective and long-lasting effect on gut microbiome composition [17]. However, the question of whether a specific taxonomic group is potentially responsible for PD development remains unresolved.

Aside from further exploring PD pathogenesis, this review will hopefully advance research into potential gut biomarkers for PD risk group assessment. Currently, a diagnosis of PD is typically made upon its motor manifestation, which occurs when an estimate of 30–60% of neurons in the substantia nigra (SN) have already been damaged [18,19]. Detecting precise PD risk biomarkers would lead to quicker diagnoses, ideally before these patients even begin to experience the motor phase of this disease. This would hopefully pave the way for earlier medical supervision and a more holistic, prevention-based approach towards individuals with prodromal PD. Additionally, implicating specific gut bacteria would help corroborate the possible effectiveness of interventional fecal microbiota transplantation (FMT) as a method of treatment in PD. If it were to be approved, it would be the first line of therapy directed against the precise cause of idiopathic PD, as currently available treatment is limited to alleviating symptoms without curbing this disease’s progression [20].

## 2. Materials and Methods

In this review, original research papers investigating gut microbiota composition in Parkinson disease were analyzed. A systematic literature search was performed according to the 2020 PRISMA updated guideline statement [21]. The review was registered in PROSPERO (ID CRD42022337225). The process was overseen by two authors (J.M.N., M.K.). The PubMed, Embase, and Scopus databases were searched on 13 June 2022, for original research articles involving human subjects. The terms “(Parkinson disease AND gut AND (bacteria OR microbiome OR microbiota)” were used for PubMed and Embase, and “(Parkinson disease AND gut AND (bacteria OR microbiome OR microbiota) AND (LIMIT-TO (SUBJAREA, “MEDI”)) AND (LIMIT-TO (DOCTYPE, “ar”)) AND (EXCLUDE (EXACTKEYWORD, “Nonhuman”) OR (EXCLUDE (EXACTKEYWORD, “Animals”)) for Scopus. Only full-text articles published in English from 2000 to 1 June 2022 were included in the final analysis.

The yielded studies were categorized into four main groups: (1) PD subjects versus healthy controls (HC), (2) markers of disease severity, (3) markers of NMS in PD, and (4) markers of PD clinical phenotypes. A further analysis of these individual groups was later conducted. It firstly involved identifying all of the bacteria evaluated in each study in each of the above-mentioned groups. Subsequently, it was determined whether these studies had established a decrease or increase of the specified bacteria for PD subjects. An overall conclusion was then reached based on an assessment of coinciding outcomes.

## 3. Results

Our search using the above-mentioned terms revealed 1674 results in the Scopus database, 990 results in the Embase database, and 551 results in the PubMed database. After an automatic (EndNote) and manual (J.M.N., M.K.) removal of duplicates, a total of 2441 papers were identified that met the search criteria. In the next step, the available abstracts were read to identify original research papers on changes in the gastrointestinal microbiota of PD patients. We excluded original papers on intestinal permeability, nasal and oral microbiomes, and interventional studies (e.g., treatment with probiotics, fecal matter transplantation). A PRISMA flow diagram of the search procedure is available as Figure 1. We identified a total of 51 original papers on 15 June 2022.

The yielded studies were then categorized into the previously mentioned groups, which revealed the following results: 38 studies in the PD subjects vs. HC group, 26 studies in the PD severity group, 17 in the NMS group, and 9 in the PD clinical phenotype group. Some studies matched into more than one of these categories.

### 3.1. Parkinson Disease Patients and Healthy Controls

Most studies investigating gut microbiota in PD chose to compare PD subjects to healthy controls. This is the preferred methodology when attempting to identify specific PD biomarkers, as it limits external, interfering factors. The full results of the previously mentioned analysis can be found as Appendix A, and a summarized version can be found below as Table 1. After careful examination of Appendix A, 22 bacteria (regardless of their taxonomic level) were deemed potentially significant, with five or more papers implicating the same bacteria, at the same taxonomic level, with the same outcome for PD subjects. Fourteen of the 22 identified bacteria were increased in PD: *Akkermansia* genus (12 studies), *Verrucomicrobiaceae* family (8 studies), *Rikenellaceae* family (8 studies), *Lactobacillus* genus (8 studies), *Lactobacillaceae* family (7 studies), *Bifidobacterium* genus (7 studies), *Bifidobacteriaceae* family (7 studies), *Proteobacteria* phylum (6 studies), *Alistipes* genus (6 studies), *Actinobacteria* phylum (6 studies), *Verrucomicrobia* phylum (6 studies), *Enterobacteriaceae* family (5 studies), *Streptococcus* genus (five studies), and *Ruminococcaceae* family (5 studies). The remaining eight of the 22 bacteria were found to be decreased in PD. These include: *Roseburia* genus (11 studies), *Lachnospiraceae* family (10 studies), *Faecalibacterium* genus (9 studies), *Prevotellaceae* family (6 studies), *Prevotella* genus (5 studies), *Blautia* genus (5 studies), *Bacteroidetes* phylum (5 studies), and *Fusicatenibacter* genus (5 studies). An additional group of bacteria, the *Desulfovibrionaceae* family, was considered potentially valuable, even though it did not meet the previously mentioned criteria. The decision to distinguish this bacterial family was based on a singular study dedicating its investigation to this specific bacterial group [22]. This study was able to reach a clear and concise conclusion of the *Desulfovibrionaceae* family’s significance as a potential PD biomarker.

### 3.2. Parkinson Disease Severity

Disease severity and/or disease duration may alter the gut microbiome in PD patients. It has been hypothesized that, throughout the course of this disease, certain bacteria may gradually change in abundance concomitantly with PD symptom progression. The identified studies on PD severity and their conclusions can be found below in Table 2. However, several differing interpretations of disease severity were found between studies, hence the discrepancy among their methodological approaches: 17 papers used the Hoehn & Yahr scale, 22 papers used UPDRS, 10 papers used Levodopa Equivalent Dose (LED), and 12 of them used disease duration for PD severity evaluations. Some studies applied more than one of the above-mentioned methods. In this category, the maximum number of coinciding outcomes attributed to a specific bacterium was three, a number held only by two bacteria: the genus *Blautia* and the genus *Bilophila*, reported separately in three papers. Both of them could potentially be of value as PD severity biomarkers. Nevertheless, further evidence on these bacteria is essential in order to reach any preliminary conclusions on their merit. Additionally, the *Desulfovibrionaceae* family was deemed potentially significant as a PD severity biomarker, even though it was not particularly distinctive in the conducted analysis (Table 2). Similarly to the PD vs. HC category, the decision to distinguish this bacterial family was based on a singular study dedicating its investigation to this specific bacterial group [22].

### 3.3. Non-Motor Symptoms

NMS of PD may be particularly burdensome for certain patients. The discovery of distinct NMS biomarkers could be a useful tool when targeting specific non-motor symptoms for treatment. In addition to this, identifying NMS biomarkers could provide new insight into PD pathogenesis. The following non-motor symptoms were examined in the identified studies: nine papers chose to focus on gastrointestinal (GI) symptoms (most commonly constipation; with the use of i.a. the Bristol and Wexner scales); four focused on cognitive decline (MoCA and MMSE scales were mostly used); one paper analyzed difficulties in daily living (using the PDQ-39 scale), body mass index (BMI), depression, and chronic pain; and one paper explicitly analyzed weight loss (WL). The results are summarized in Table 3. After careful examination of Table 3, the authors were not able to identify a single potential biomarker, as the number of papers on individual NMS was not sufficient in order to reach a clear, concise conclusion. Additionally, some studies generated contradictory outcomes. *Akkermansia* was found to be elevated in constipated PD patients in two studies [49,58].

### 3.4. Clinical Phenotype

The pathophysiology of individual PD clinical subtypes is yet to be resolved. However, some studies have been able to identify a link between distinct bacteria and PD motor phenotype. The results of the previously mentioned analysis of studies investigating clinical phenotypes can be found in Table 4. The following clinical phenotypes were evaluated in the selected studies: one study examined both early-onset PD and late-onset PD; one study distinguished non-tremor PD; four focused on tremor-dominant PD; two studies focused on postural instability and gait disorders (PIGD); one examined dyskinetic PD; and finally one study analyzed hypokinetic-rigid PD. After careful examination of Table 4, the authors were not able to identify any potential biomarkers, as the number of papers on individual clinical phenotypes of PD was not sufficient to reach a clear, concise conclusion.

## 4. Discussion

Microbiological research on PD pathophysiology is rapidly gaining momentum. In contrast to other neurodegenerative diseases involving the CNS, early involvement of the peripheral nervous system is a unique, yet somewhat puzzling, element of PD [8,9,67]. The role that bacteria have in triggering intestinal inflammation is currently being extensively investigated, with research proposing a handful of mechanisms. These include the following: (1) a disruption of the mucus layer in the intestine (*Akkermansia muciniphila*, *Bifidobacterium*, *Desulfovibrionaceae*), (2) a disruption in short-chained fatty acid (SCFA) production, (3) an increased production of proinflammatory cytokines (TNF, IL-1, IL-17, IFN-gamma, and IL-6), and (4) lipopolysaccharide (LPS) production in the intestine [22,68,69]. Microbial metabolites and their pathophysiological link to PD are an especially important and constantly growing area of research. It has been established that stool and serum bacterial metabolites, along with inflammatory cytokines, have an influence on glia maturation and functioning [70]. Substances such as LPS or SCFA play vital roles in the complex process that is neuroinflammation [69,71,72]. Mucosal layer disturbances caused by bacteria and their byproducts, in conjunction with intestinal barrier integrity dysfunction, may lead to an increased enteric nervous system exposure to high amounts of toxins [15]. Elevated levels of fecal calprotectin (a marker of intestinal inflammation), fecal alpha-1-antitrypsin, and fecal zonulin (both markers of intestinal permeability) in PD patients further support this hypothesis [73].

Identifying a distinct microbe involved in PD pathogenesis would be groundbreaking, as it could initiate a shift from the current symptomatic treatment of PD to more causative and targeted therapies. One of the main limitations of this review is that the currently available studies apply differing approaches towards microbiome analysis, both in methodology and nomenclature. Regarding methodology, the studies yielded in this review used either 16s rRNA or 16s rDNA for microbiological analyses, with some additionally analyzing bacterial metabolites. As for nomenclature, different taxonomic levels of bacteria were examined, which led to some difficulties when searching for coinciding results among studies. It cannot be presumed, for example, that an increase of a specific species reported in one study is equivalent to an increase of the whole taxonomic family to which this species belongs, and vice versa.

Microbiota studies are often difficult to interpret due to a great diversity of obtained results. Numerous external factors such as diet, circadian rhythm, concomitant diseases, and medication have an enormous influence on gut microbiota [74,75]. The results of our literature review are partially in line with a recent meta-analysis conducted by Toh et al. [76]. The authors analyzed raw 16s rRNA sequences from 10 studies, which included a total of 969 PD patients and 734 controls. They established that factors, such as race (Caucasian vs. non-Caucasian) and geographical location, influence intestinal microbiota composition. They reported elevated levels of *Akkermansia* and *Hungatella* and reduced levels of *Roseburia* and *Faecalibacterium* in PD subjects. Interestingly, the authors determined that a reduced abundance of *Roseburia* and unclassified *Lachnospiracea* was identified in both Caucasian and non-Caucasian cohorts. The results of this important study suggest that a specific pathogens may be involved in PD pathogenesis, irrespective of race.

A reduced abundance of the *Roseburia* species in PD, which has been reported in our review as well as in a recent metanalysis, is of particular interest. This species is considered “a biomarker of good health” [1]. Its decreased levels were described not only in PD, but also in both depressive and bipolar disorders, and in Alzheimer disease [77,78]. These commensal bacteria play an important role in the production of colonic butyrate, which is considered to be one of the most important SCFAs [79]. Lower levels of *Roseburia* and certain serum SCFAs in PD subjects are most likely correlated with one another.

A study explicitly focusing on the *Desulfovibrionaceae* family is particularly noteworthy, as it clearly establishes the importance of the *Desulfovibrionaceae* in PD pathogenesis [22]. According to Murros et al., members of this bacterial family adhere to the intestinal wall whilst producing LPS and hydrogen sulfide, a chemical considered neurotoxic in high concentrations. It has been proposed that hydrogen sulfide may trigger alpha-synuclein aggregation, leading to intestinal neurodegeneration. Bacteria from the *Desulfovibrionaceae* family, when present in disproportionate amounts, disrupt butyrate production [22]. This unique set of features, combined with the corroborating results of the above-mentioned study, explicitly implicate the involvement of the *Desulfovibrionaceae* in PD pathogenesis.

Another important aspect of microbial PD research is the heterogeneity of the disease itself. It has been postulated that PD should be considered a syndrome rather than a homogeneous disease, due to its varied manifestation among PD patients [80]. Although it is possible that patients with different PD phenotypes share specific microbiome signatures, the results from the studies investigating distinct PD subtype pathogens are incompatible with one another [33,34,37,41,43,58,66].

Finally, it cannot be presumed that the microbial trigger potentially involved in PD pathogenesis is present throughout the whole course of the disease. It may reside in the intestine only for a limited amount of time, perhaps even years before motor symptom onset. If this were to be the case, the microbial culprit would be undetectable at the time of stool sample collection. Moreover, constipation, a common non-motor symptom of PD, may have a significant impact on the diverse results achieved in the conducted research [48]. It has been well-documented that constipation greatly influences gut microbiota in non-PD subjects [80,81]. It is therefore crucial that constipation be considered an important modifying factor when analyzing gut bacteria composition. The question remains whether changes in gut microbiota are the cause or the consequence of altered peristaltic movements in PD subjects [82].

## 5. Conclusions

We conclude that future research on gut microbiota in PD subjects should be comprised of large, international data sets, preferably with patients of varying ethnic backgrounds and residing in different geographical locations. This would subdue the interference of local factors, such as diet. Nevertheless, recent studies have already been able to reach some interesting and valuable preliminary conclusions, which may be helpful in determining future directions of research in this field. The findings deemed most significant in this review were elevated levels of the *Akkermansia* genus, the *Verrucomicrobiaceae* family, the *Rikenellaceae* family, and the *Lactobacillus* genus in PD subjects. These bacteria could potentially be used as PD biomarkers. In addition to this, reduced levels of the *Roseburia* genus and the *Lachnospiraceae* family in PD subjects were also noteworthy; we encourage that they be further investigated in the future for their potentiality as biomarkers. Each of the above-mentioned bacteria, which have been highlighted by previous studies of gut microbiota in PD, could possibly serve as a target for prospective microbiota-modifying therapies, such as probiotic treatment or fecal matter transplantation. Moreover, it is our recommendation that detailed clinical characteristics of PD patients be recorded, as this step is crucial for distinguishing distinctive bacterial fingerprints. Unified nomenclatures of microbial taxonomic levels should be adopted between studies, as it would allow for more precise and transparent conclusions when comparing their outcomes.

## Figures and Tables

**Figure 1 biomedicines-10-02057-f001:**
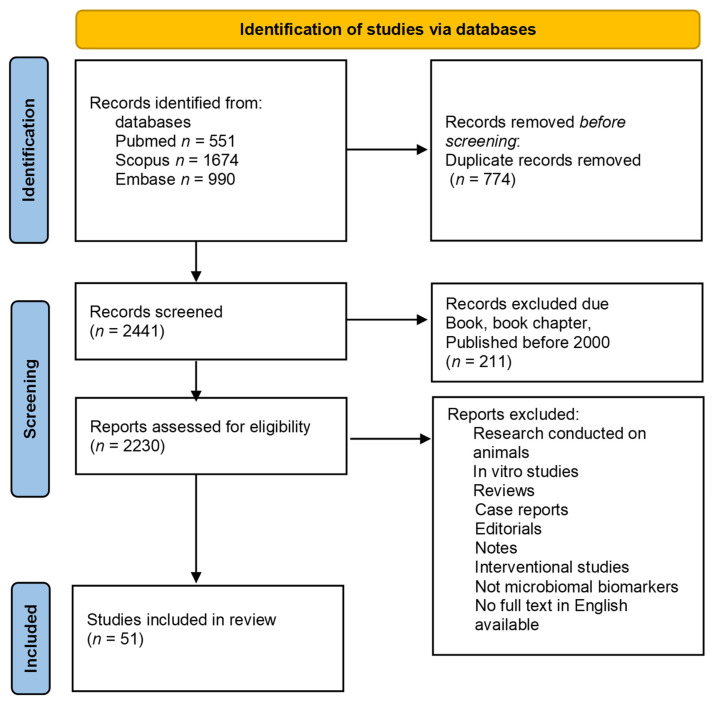
PRISMA flowchart in accordance with the PRISMA 2020 statement. (*n*—number).

**Table 1 biomedicines-10-02057-t001:** Parkinson disease patients and healthy controls: research papers and their conclusions—summarized version ([↑/↓]-increase/decrease of abundance in comparison patients with Parkinson’s Disease to Healthy Control, PD—patients with Parkinson’s Disease, HC—healthy controls).

Taxonomic group	Item	(Li, Wang et al., 2019) [23]	(Li, Lu et al., 2021) [24]	(Kenna, Chua et al., 2021) [25]	(Li, Cui et al., 2019) [26]	(Kim, Jung et al., 2022) [27]	(Keshavarzian, Green et al., 2015) [28]	(Li, Wu et al., 2017) [29]	(Lin, Zheng et al., 2018) [30]	(Qian, Yang et al., 2020) [31]	(Petrov, Saltykova et al., 2017) [32]	(Pietrucci, Cerroni et al., 2019) [33]	(Lin, Chen et al., 2019) [34]	(Yan, Yang et al., 2021) [35]	(Zhang, Yue et al., 2020) [36]	(Scheperjans, Aho et al., 2015) [37]	(Vascellari, Palmas et al., 2020) [38]	(Qian, Yang et al., 2018) [39]	(Ren, Gao et al., 2020) [40]	(Weis, Schwiertz et al., 2019) [41]	(Wallen, Appah et al., 2020) [42]	(Aho, Pereira et al., 2019) [43]	(Baldini, Hertel et al., 2020) [44]	(Barichella, Severgnini et al., 2019) [45]	(Bedarf, Hildebrand et al., 2017) [46]	(Chen, Chen et al., 2022) [47]	(Chen, Bi et al., 2021) [48]	(Cirstea, Yu et al., 2020) [49]	(Cosma-Grigorov, Meixner et al., 2020) [50]	(Del Chierico, Grassini et al., 2020) [51]	(Heintz-Buschart, Pandey et al., 2018) [52]	(Hill-Burns, Debelius et al., 2017) [53]	(Lubomski, Xu et al., 2022) [54]	(Lubomski, Xu et al., 2022) [55]
Phylum	*Actinobacteria*		↑					↑						↑	↑		↑		↑															
	*Bacteroidetes*					↓	↑	↓						↑	↓		↓		↓															
	*Firmicutes*						↓		↓					↓	↑										↑									
	*Proteobacteria*			↑			↑	↑									↑							↑						↑				
	*Verrucomicrobia*				↑		↑			↑			↑		↑									↑										
Family	*Bifidobacteriaceae*		↑						↑					↑			↑					↑		↑								↑		
	*Enterobacteriaceae*			↑				↑				↑	↑											↑									↑	
	*Lachnospiraceae*			↓			↓		↓			↓					↓					↓		↓				↓		↓		↓		
	*Lactobacillaceae*		↑	↓	↓							↑	↑			↑		↓				↑		↑								↑	↑	↑
	*Prevotellaceae*	↓											↓			↓						↓			↓					↓				
	*Rikenellaceae*		↑		↑								↑	↑			↑	↑	↑			↑												
	*Ruminococcaceae*	↑			↑											↑			↑										↑					
	*Verrucomicrobiaceae*				↑		↑						↑			↑	↑							↑	↑						↑	↑		
Genus	*Akkermansia*	↑			↑		↑						↑		↑		↑						↑	↑	↑			↑			↑	↑		
	*Alistipes*		↑		↑					↑				↑				↑	↑															
	*Bifidobacterium*		↑								↓			↑			↑				↑	↑						↑				↑	↑	
	*Blautia*						↓	↓									↓					↓										↓		
	*Fusicatenibacter*																			↓	↓						↓						↓	↓
	*Faecalibacterium*							↓			↓		↓	↓						↓	↓							↓	↓			↓	↑	
	*Lactobacillus*		↑		↓					↑	↑		↑					↓			↑	↑	↑									↑		
	*Oscillospira*						↑				↑				↑						↓			↓										
	*Parabacteroides*				↑								↑	↑			↓							↑										
	*Prevotella*					↓					↓		↓								↑	↓			↓									
	*Roseburia*		↓	↓	↑		↓					↓					↓				↓	↓		↓				↓		↓		↓		
Size of Groups	PD	10	69	87	51	10	38	24	75	144	89	80	80	20	63	72	56	45	27	34	520	64	147	193	31	96	29	197	71	20	76	197	103	74
	HC	10	244	47	39	10	34	14	45	141	66	72	77	20	137	72	0	45	13	25	314	64	162	113	28	85	15	103	30	8	78	130	81	74

**Table 2 biomedicines-10-02057-t002:** Parkinson disease severity: research papers and their conclusions (H&Y—Hoehn and Yahr Scale, UPRDS—Unified Parkinson’s Disease Rating Scale, LED—Levodopa Equivalent Dose, UPRDS-III—Unified Parkinson’s Disease Rating Scale 3rd part, DD— disease duration, PD—patient with Parkinson’s disease, HC—healthy control, DC—diseased control).

Title of Original Article	Parameters Rated in Original Article	Groups’ Size	Microbiota Changes
Murros, Huynh et al., 2021 [22]	H&Y	20 PD, 20 HC	*Desulfovibro* positively correlated with progression
Jin, Li et al., 2019 [56]	H&Y, UPRDS, DD	72 PD, 68 HC	(1) *Eubacterium* positively correlated with severity (H&Y and UPRDS-III) (2) *Prevotella* and *Lachnospira* negatively correlated with severity (H&Y and UPRDS-III)
Li, Cui et al., 2019 [26]	UPRDS	51 PD, 39 HC	(1) *Ruminococcus torques* positively correlated with UPRDS (2) *Bacillales* and *Pseudomonas veronii* negatively correlated with UPRDS
Keshavarzian, Green et al., 2015 [28]	H&Y, UPRDS, DD	38 PD, 34 HC	(1) *Bacteroidetes* and *Proteobacteria* positively correlated with DD (2) *Firmicutes*, *Lachnospiraceae* and *Blautia* negatively correlated with DD
Li, Wu et al., 2017 [29]	H&Y, UPRDS, DD	24 PD, 14 HC	(1) *Enterococcus, Proteus* and *Escherichia-Shigella* positively correlated with UPRDS and DD, *Megasphaera* positively correlated with DD (2) Blauti, *Ruminococcus* and *Haemophilus* negatively correlated with UPRDS nad DD, *Faecalibacterium* and *Odoribacter* negatively correlated with UPRDS and *Sporobacter* negatively correlated with DD
Lin, Zheng et al., 2018 [30]	DD	75 PD, 45 HC	*Rikenellaceae, Deferribacteraceae* and *Deferribacteraceae* positively correlated with DD
Minato, Maeda et al., 2017 [57]	H&Y, UPRDS, LED	36 PD	*Bifidobacterium* and *Atopobium* positively correlated with UPRDS
Pietrucci, Cerroni et al., 2019 [33]	H&Y, UPRDS	80 PD, 72 HC	*Lachnospiraceae* and *Enterobacteriaceae* negatively correlated with H&Y and UPRDS
Khedr, Ali et al., 2021 [58]	H&Y, UPRDS, DD	46 PD, 31 HC	*Bifidobacterium* negatively correlated with DD
Zhang, Yue et al., 2020 [36]	H&Y, DD	63 PD, 137 HC	(1) *Methanobrevibacter, Eggerthella, Akkermansia, Adlercreutzia, Collinsella, Coprococcus* and *Parabacteroides* positively correlated with H&Y and DD, *Desulfovibrio, Holdemania, Pyramidobacter*, *Anaerostipes* and *Acidaminococcus* positively correlated with H&Y, *Blautia* positively correlated with DD (2) *Paraprevotella* and *Fusobacterium* negatively correlated with H&Y and *Bifidobacterium* and *Roseburia* negatively correlated with DD
Takahashi, Nishiwaki et al., 2022 [59]	UPRDS, LED, DD	223 PD	(1) *Bifidobacterium, Bilophila, Lactobacillus, Oscillibacter, Tyzzerella* and *Lachnospiraceae NK4A136* positively correlated with W/O, *Pediococcus* and *Alloprevotella* positively correlated with dyskinesia (2) *Blautia* negatively correlated with W/O, dyskinesia, DD and LED, *Anaerostipes*, *Fusicatenibacter* and *Lachnospiraceae Eligens group* negatively correlated with W/O and *Eggerthella* negatively correlated with dyskinesia
Scheperjans, Aho et al., 2015 [37]	H&Y, UPRDS	72 PD, 72 HC	*Enterobacteriaceae* positively correlated with COMT inhibitors intake
Rosario, Bidkhori et al., 2021 [60]	UPRDS, DD	26 PD, 11 HC 14 DC	*Escherichia Coli, Erysipelatoclostridium sp1* and *Victivallis vadensis* positively correlated with UPRDS
Qian, Yang et al., 2018 [39]	H&Y, UPRDS, LED, DD	45 PD, 45 HC	*Escherichia-Shigella* and *Comamonas* negatively correlated with DD, *Intestinimonas, Dorea* and *Phascolarctobacterium* negatively correlated with LED *Saccharibacteria genera incerta sedi* negatively negatively correlated with UPRDS
Weis, Schwiertz et al., 2019 [41]	H&Y	34 PD, 25 HC	*Faecalibacterium* and *Peptoniphilus* positively correlated with H&Y
Aho, Pereira et al., 2019 [43]	UPRDS, LED	64 PD, 64 HC	*Prevotella* negatively correlated with disease progression
Baldini, Hertel et al., 2020 [44]	H&Y, UPRDS, LED	147 PD, 162 HC	(1) *Bilophila* positively correlated with H&Y, *Peptococcus* and *Flavonifactor* positively correlated with UPRDS-III (2) *Paraprevotella* negatively correlated with UPRDS-III and H&Y
Barichella, Severgnini et al., 2019 [45]	UPRDS	193 PD, 113 HC	*Lactobacillus* positively correlated with UPRDS-III and postural instability
Bedarf, Hildebrand et al., 2017 [46]	UPRDS	31 PD, 28 HC	No significant taxonomic associations were detected, neither at genus nor at species level, when microbiota abundance was correlated with clinical data
Chen, Bi et al., 2021 [48]	H&Y, UPRDS	29 PD, 15 HC	No significant taxonomic associations were detected, neither at genus nor at species level, when microbiota abundance was correlated with clinical data
Cilia, Piatti et al., 2021 [61]	H&Y, UPRDS, LED	39 PD	(1) *Lactobacillaceae* postively correlated with faster progression in H&Y (2) *Fusobacterium* negatively correlated with faster progression in H&Y
Heintz-Buschart, Pandey et al., 2018 [52]	H&Y, UPRDS	76 PD, 78 HC	*Akkermansia, Anaerotruncus* spp. and *Clostridium XIVa* positively correlated with UPRDS
Nishiwaki, Ito et al., 2022 [62]	UPRDS, LED, DD	104 PD	*Akkermansia* positively correlated with faster progresion, *Fusicatenibacter*, *Faecalibacterium* and *Blaustia* negatively correlated with faster progresion
Zhang, He et al., 2022 [63]	UPRDS, LED, DD, H&Y	106 PD	*Subdoligranulum* positively correlated with progression of PD, *Lachnospirales*, *Lachnospiraceae*, *Burkholderiales*, *Parasutterella*, *Desulfobacterota*, *Desulfovibrionales*, *Desulfovibrionaceae* negatively corelated with progresion
Lubomski, Xu et al., 2022 [54]	LED, UPRDS, H&Y, DD	103 PD, 81 HC	*Enterobacteriales*, *Lactobacillaceae*, *Enterobacteriaceae* and *Enterococcaceae* positively correlated with LED, *Lactobacillales* positively correlated with UPRDS. *Lactobacillaceae*, *Enterobacteriaceae*, *Escherichia/Shigella*, *Lactobacillus* and *Eggerthella* positively correlated with H&Y
Lubomski, Xu et al., 2022 [55]	UPRDS, LED	74 PD, 74 HC	*Barnesiella* and *Barnesiellaceae* negatively correlated with faster progresion

**Table 3 biomedicines-10-02057-t003:** Non-motor symptoms of Parkinson disease: research papers and their conclusions (PD—patients with Parkinson’s disease, HC—healthy controls, C—constipation, WS—Wexner score, NMSQ—Non-Motor Symptoms Questionnaire, PDQ39—The Parkinson’s Disease Questionnaire, MoCA—Montreal Cognitive Assessment, MMSE—Mini-mental state examination, NMS—non-motor symptoms, IBS—Irritable bowel syndrome).

Title of Original Article	Groups’ Size	Microbiota Changes
Mertsalmi, Aho et al., 2017 [64]	74 PD, 75 HC	*Prevotellaceae, Prevotella* and *Bacteroides* negatively correlated with IBS-like symptoms
Murros, Huynh et al., 2021 [22]	20 PD, 20 HC	*Desulfovibrio* positively correlated with constipation
Kenna, Chua et al., 2021 [25]	87 PD, 47 HC	No significant taxa
Qian, Yang et al., 2020 [31]	144 PD, 141 HC	No significant taxa
Khedr, Ali et al., 2021 [58]	46 PD, 31 HC	*Akkermansia* positively correlated with constipation
Rosario, Bidkhori et al., 2021 [60]	26 PD, 25 HC	*Erysipelatoclostridium, Escherichia Coli* and *Methanobrevibacter smithii 1* positively correlated with GI dysfunction
Qian, Yang et al., 2018 [39]	45 PD, 45 HC	*Methanobrevibacter smithii 1* and *Clostridium XlVb* negatively correlated with MMSE
Ren, Gao et al., 2020 [40]	27 PD, 13 HC	*Butyricimonas* negatively correlated with MMSE and MoCa, *Desulfovibrio*, *Ruminococcus, Bilophila, Barnesiella, Acidaminococcus, Pyramidobacter and Oxalobacter* negatively correlated with MoCa and *Sutterella, Alistipes, Odoribacter, Hungatella, Helicobacter, Solobacterium, Oscillospira* and *Hydrogenoanaerobacterium* negatively correlated with MoCa
Baldini, Hertel et al., 2020 [44]	147 PD, 162 HC	*Bifidobacterium* positively correlated with constipation
Barichella, Severgnini et al., 2019 [45]	193 PD, 113 HC	*Christensenellaceae* and *Lactobacillaceae* positively correlated with intelectual impairment and *Lachnospiraceae* and *Lactobacillus* negatively correlated with intelectual impairment
Chen, Bi et al., 2021 [48]	29 PD, 15 HC	(1) *Veillonella* positively correlated with WS, PDQ39 and NMSQ, *Hungatella, Streptococcus* and *Anaerrotruncus* positively correlated with WS and C, *Actinobacteria, Coriobacteriia, Coriobacteriaceae, Streptococcaceae, Subdoligranulum sp_4_3_54A2FAA* and *Streptococcus salivarius sb. Salivarius* positively correlated with C and *Sellimonas* and *Faecalitalea* positively correlated with WS (2) *Holdemanella* and *Megamonas* negatively correlated with WS and C and *Sutterella parviubra* and *Prevotella stercorea* negatively correlated with C
Cilia, Piatti et al., 2021 [61]	39 PD	*Oscillospira* positively correlated with MMSE i MoCA deterioration rate, *Actinobacteria* positively correlated with MMSE deterioration rate, *Roseburia* and *Faecalibacterium* negatively correlated with MMSE deterioration rate & NMS progression and *Lachnospiraceae* negatively correlated with NMSQ (GI and behavioral parts)
Cirstea, Yu et al., 2020 [49]	197 PD, 103 HC	*Faecalibacterium*, *Dorea*, *Oscillospira* and *Ruminococcus* positively correlated with C, *Roseburia* and *Prevotella* positively correlated with Bristol scale and *Akkermansia*, *Christensenellaceae*, *Ruminococcus*, *Oscillospira* and *Dorea* negatively correlated with Britstol scale
Del Chierico, Grassini et al., 2020 [51]	20 PD, 8 HC	*Christensenellaceae*, *Clostridiaceae*, *Peptococcaceae*, *Desulfovibrionaceae*, *Ruminococcaceae*, *Roseburia*, *Lachnospira*, *Ruminococcus bromii* and *Collinsella aerofaciens* positively correlated with weight loss, *Streptococcaceae, Enterobacteriaceae*, *Enterococcus*, *Escherichia Coli*, *Enterobacter cecorum* and *Faecalibcterium prausnitzii* negatively correlated with weight loss and *Eubacterium lenta* negatively correlated with weight loss and MoCA
Heintz-Buschart, Pandey et al., 2018 [52]	76 PD, 78 HC	No significant taxa
Lubomski, Xu et al., 2022 [54]	103 PD, 81 HC	*Pseudoflavonifractor* positively correlated with NMS, *Gordonibacter*, *Eggerthella* and *Pseudoflavonifractor* positively correlated with PDQ-39, *Veillonella*, *Klebsiella* and *Pseudoflavonifractor* positively correlated with depression, *Enterobacteriaceae* and *Veillonella* negatively correlated with chronic pain, *Bacteroidaceae* and *Synergistaceae* positively correlated with chronic pain, *Butyricoccus*, *Faecalibacterium* and *Coprococcus* positively correlated with Bristol score, *Escherichia*/*Shigella* negatively correlated with Bristol score, *Holdemania* and *Butyricicoccus* positively correlated with ROME-IV score, *Romboutsia* negatively correlated with ROME-IV score, *Romboutsia* positively correlated with Body Mass Index, *Anaeroplasma* negatively correlated with Body Mass Index
Fu, Shih et al., 2022 [65]	199 PD, 131 HC	*Anaerostipes*, *Fusicatenibacter*, *Lachnospiraceae ND3007 group, Blautia*, *Ruminococcaceae UCG 013*, *Coprococcus 3*, *Faecalibacterium*, *Dorea*, *Lachnospiraceae UCG 004*, *Anaerostipes hadrus*, *Fusicatenibacter saccharivorans*, *Coprococcus 3* comes and *Dorea longicatena* positively correlated with constipations

**Table 4 biomedicines-10-02057-t004:** Clinical phenotypes of Parkinson disease: research papers and their conclusions (TD—tremor-dominant phenotype of Parkinson’s disease, NTD—non-tremor dominant phenotypes of Parkinson’s disease, PD—patient with Parkinson’s disease, HC—healthy control, PIGD—Postural Instability and Gait Disorder).

Title of Original Article	Groups’ Size	Microbiota Changes
Lin, Zheng et al., 2018 [30]	75 PD, 45 HC	*Pasteurellaceae, Alcaligenaceae, Lactococcus, Facklamia, Clostriadium, Sutterella, Faecalibacterium, Leptotrichia, Haemophilus, Comamonas* and *Anaerotruncus* increased in early-onset PD, *Leptotrichia* increased in TD, *Roseburia* increased in NTD
Pietrucci, Cerroni et al., 2019 [33]	80 PD, 72 HC	No significant differences were observed among the different PD phenotypes.
Lin, Chen et al., 2019 [34]	80 PD, 77 HC	*Firmicutes, Verrucomicrobia, Clostridia, Verrucomicrobiae, Clostridiales, Verrucomicrobiales* and *Verrucomicrobiaceae* positively correlated with TD, *Bacteroidetes, Bacteroidia, Bacteroidales, Alcaligenaceae, Sutterella, Mogibacterium, Flavobacterium, Desulfovibro, Propionibacterium* and *Cupriavidus* positively correlated with NTD
Khedr, Ali et al., 2021 [58]	46 PD, 31 HC	No significant differences of gut microbiota between tremor-dominant, akinetic-rigid, and mixed types of PD
Scheperjans, Aho et al., 2015 [37]	72 PD, 72 HC	*Enterobacteriaceae* increased in PIGD
Vascellari, Melis et al., 2021 [66]	56 PD	(1) *Lactobacillaceae and Lactobacillus* increased in D and *Brevibacteriaceae, Clostridiaceae, Gemellaceae, Lachnospiraceae, Faecalibacterium, Brevibacterium, Tindallia, Gemella, Blautia, Coprococcus* and *Lachnospira* increased in TD (2) *Enterobacteriaceae, Serratia* and *Escherichia* decreased in TD
Weis, Schwiertz et al., 2019 [41]	34 PD, 25 HC	(1) *Peptoniphilus* increased in hypokinetic-rigid PD (2) *Faecalibacterium* decreased in hypokinetic-rigid PD and *Ruminococcaceae* decreased in TD
Lubomski, Xu et al., 2022 [54]	103 PD, 81 HC	*Synergistaceae*, *Oxalobacteraceae* and *Faecalicoccus* increased in PIGD
Aho, Pereira et al., 2019 [43]	64 PD, 64 HC	No significant difference

## Data Availability

All data used in this study are available in Appendix A.

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
