# Peer review of "Microbiota Dysbiosis in Parkinson Disease—In Search of a Biomarker"

_biomedicines, 2022, doi:10.3390/biomedicines10092057_

Round 1

Reviewer 1 Report

Biomedicines-1812922

Full Title: Microbiota dysbiosis in Parkinson disease - in search of a biomarker

The main objective of this work is the review of previously published works to identify a potential gut bacterial biomarkers of Parkinson disease.

The authors discuss several possible bacterial candidates that are present (as Desulfovibrionaceae and others), and in some cases absent (Roseburia), but do not come to a firm conclusion about a possible biomarker.

In general, this work is original, well structured, easy to read, and without gross methodological errors, but I'd like to make the following considerations/questions to the authors:

1.- Review the superscript for the author work Department. Either separate, as it is on the first line, or attached to the department name, as it is on the second line.

2.- The asterisk is normally used to indicate the corresponding author.

3.- Line 17: vs is the abbreviation of versus, a Latin word that must be written in italics and, therefore, its abbreviation as well.

4.- Lines 68 and 69: check the sentence “diagnosis of PD is typically made upon its motor manifestation, which occurs when an estimate of 30% of neurons in the substantia nigra (SN) have already been damaged” Only 30%?

5.- Line 93 and others: If NMS is “non-motor symptoms” typing “NMS symptoms” is redundant. Please, suppress symptoms.

6.- Table 2, line 2: correct “Desulfvibro”

7.- Line 175 and others: indicate the numbers with letters, nine papers …, four focused … etc.

In conclusion, this is a very interesting work can be improved by some little changes.

Author Response

Thank you for reviewing our paper. Below you will find are our answers regarding your suggestions:

1) Review the superscript for the author work Department. Either separate, as it is on the first line,
or attached to the department name, as it is on the second line.
2) The asterisk is normally used to indicate the corresponding author.
3) Line 17: vs is the abbreviation of versus, a Latin word that must be written in italics and,
therefore, its abbreviation as well.
Points 1), 2) and 3) have been corrected in the text as suggested.

4) Lines 68 and 69: check the sentence “diagnosis of PD is typically made upon its motor
manifestation, which occurs when an estimate of 30% of neurons in the substantia nigra (SN)
have already been damaged” Only 30%?
Thank you for this remark, we have changed this part of the sentence to 30%-60%, in accordance with
our citations.

5) Line 93 and others: If NMS is “non-motor symptoms” typing “NMS symptoms” is redundant.
Please, suppress symptoms.
6) Table 2, line 2: correct “Desulfvibro”
7) Line 175 and others: indicate the numbers with letters, nine papers …, four focused … etc.
Points 5), 6) and 7) have been corrected as suggested.

Reviewer 2 Report

Dear Authors,

I have read the manuscript with inetrest and I send you my comments:

1) It could be useful to add a table with the differentiation for age and a table with a differentiation for sex

2) In the title you write "- in search of a biomarker", please refer what may be the biomarker.

You report the role of bacteria, but you don't report the role of biomarker, plase add

Author Response

Thank you for reviewing our paper. Below you will find are our answers regarding your suggestions:
1) It could be useful to add a table with the differentiation for age and a table with a differentiation
for sex
Thank you for this remark. We agree that sex and age may influence microbiota composition.
However, we did not find this information provided in every paper reviewed. Also, we believe it
could make the data exceptionally difficult to interpret.
2) In the title you write "- in search of a biomarker", please refer what may be the biomarker. You
report the role of bacteria, but you don't report the role of biomarker, plase add

Thank you for your comment. We have expanded upon this subject in the conclusions section as
follows.
‘The findings deemed most significant in this review were elevated levels of the Akkermansia
genus, the Verrucomicrobiaceae family, the Rikenellaceae family, and the Lactobacillus genus in
PD subjects. These bacteria could potentially be used as PD biomarkers. In addition to this,
reduced levels of the Roseburia genus and the Lachnospiraceae family in PD subjects were also
noteworthy, we encourage that they be further investigated in the future for their potentiality
as biomarkers.’
In our review we implicate bacteria with the most consistent changes reported in the literature.
We believe they could be potentially used as biomarkers of Parkinson disease.

Round 2

Reviewer 2 Report

Dear authors

I have read the revised version and I have not other comments